# Global transcriptional analysis of *Geobacter sulfurreducens gsu1771* mutant biofilm grown on two different support structures

**Juan B. Jaramillo-Rodríguez**[1], **Leticia Vega-Alvarado**[2], **Luis M. Rodríguez-Torres**[1], **Guillermo A. Huerta-Miranda**[1], **Alberto Hernández-Eligio**[1,3]*, **Katy Juarez**[1]*

**1** Departamento de Ingeniería Celular y Biocatálisis, Instituto de Biotecnología Universidad Nacional Autónoma de México, Cuernavaca, Morelos, México, **2** Instituto de Ciencias Aplicadas y Tecnología, Universidad Nacional Autónoma de México, Ciudad Universitaria, Ciudad de México, México, **3** Investigador por México, Consejo Nacional de Ciencia y Tecnología, Ciudad de México, México

* katy.juarez@ibt.unam.mx (KJ); alberto.hernandez@ibt.unam.mx (AH-E)

## Abstract

Electroactive biofilms formation by the metal-reducing bacterium *Geobacter sulfurreducens* is a step crucial for bioelectricity generation and bioremediation. The transcriptional regulator GSU1771 controls the expression of essential genes involved in electron transfer and biofilm formation in *G. sulfurreducens*, with GSU1771-deficient producing thicker and more electroactive biofilms. Here, RNA-seq analyses were conducted to compare the global gene expression patterns of wild-type and Δ*gsu1771* mutant biofilms grown on non-conductive (glass) and conductive (graphite electrode) materials. The Δ*gsu1771* biofilm grown on the glass surface exhibited 467 differentially expressed (DE) genes (167 upregulated and 300 downregulated) versus the wild-type biofilm. In contrast, the Δ*gsu1771* biofilm grown on the graphite electrode exhibited 119 DE genes (79 upregulated and 40 downregulated) versus the wild-type biofilm. Among these DE genes, 67 were also differentially expressed in the Δ*gsu1771* biofilm grown on glass (56 with the same regulation and 11 exhibiting counterregulation). Among the upregulated genes in the Δ*gsu1771* biofilms, we identified potential target genes involved in exopolysaccharide synthesis (*gsu1961-63*, *gsu1959*, *gsu1972-73*, *gsu1976-77*). RT-qPCR analyses were then conducted to confirm the differential expression of a selection of genes of interest. DNA-protein binding assays demonstrated the direct binding of the GSU1771 regulator to the promoter region of *pgcA*, *pulF*, *relA*, and *gsu3356*. Furthermore, heme-staining and western blotting revealed an increase in *c*-type cytochromes including OmcS and OmcZ in Δ*gsu1771* biofilms. Collectively, our findings demonstrated that GSU1771 is a global regulator that controls extracellular electron transfer and exopolysaccharide synthesis in *G. sulfurreducens*, which is crucial for electroconductive biofilm development.

**Data Availability Statement:** All RNAseq data files are available from the Omnibus database (accession number GSE223184).

**Funding:** Our work was supported by Programa de Apoyo a Proyectos de Investigación e Innovación Tecnológica de la Universidad Nacional Autónoma de México (Grant IN212022). GAH-M is supported by Consejo Nacional de Ciencia y Tecnología postdoctoral fellowship (2322131). KJ received financial support from Programa de Apoyos para la Superación del Personal Académico de la Universidad Nacional Autónoma de México during a sabbatical stay. THE FUNDERS HAD NO ROLE IN STUDY DESIGN, DATA COLLECTION, AND ANALYSIS, DECISION TO PUBLISH, OR PREPARATION OF THE MANUSCRIPT.

**Competing interests:** The authors have declared that no competing interests exist.

# Introduction

*Geobacter sulfurreducens* is a gram-negative, anaerobic bacterium that inhabits subsurface environments. This bacterium is known for its ability to degrade organic matter with the reduction of extracellular electron acceptors such as Fe(III) and Mn(IV) oxides, U(VI), and electrodes [1]. Extracellular electron transfer (EET) is a biological process present in many bacteria, which plays a crucial role in a wide variety of physiological and environmental processes. In *G. sulfurreducens*, EET is a process driven by a repertoire of more than 100 *c*-type cytochromes and electrically conductive nanowires [2,3]. *G. sulfurreducens* has become a prominent model for studies on electricity production in bioelectrochemical systems due to its ability to directly transfer electrons to electrodes and form metabolically active biofilms, which enable the conversion of organic matter (e.g., acetate) into electricity [1,4]. In addition to participating in the recycling of organic matter present in the environment, *G. sulfurreducens* has been effectively applied to the bioremediation of subsurface environments contaminated with organic compounds and metals [2].

Biofilms are microbial communities attached to a biotic or abiotic surface, which are embedded in an extracellular matrix of polymeric substances that are synthesized by the microorganisms themselves [5]. The extracellular matrix of *G. sulfurreducens* biofilms is primarily composed of proteins and exopolysaccharides [6]. These proteins include *c*-type cytochromes, structural components and electrically conductive nanowires which also have an adherent role in biofilm structure [7,8]. Furthermore, exopolysaccharides enable cell agglutination and adherence to abiotic surfaces, in addition to anchoring *c*-type cytochromes to the extracellular matrix [9,10]. *G. sulfurreducens* strains unable to synthesize these elements develop thin biofilms that are less electroconductive than those produced by their wild-type counterparts [9,11].

The transcriptional regulator GSU1771 was first discovered through adaptive evolution experiments that enhanced the reduction of Fe(III) oxides in *G. sulfurreducens* [12]. The *gsu1771* gene encodes a transcriptional regulator of the SARP family C-terminal region, which exhibits a winged helix-turn-helix (HTH) DNA-binding motif, a central activator domain, and a response receiver domain in the N-terminal. SARP family transcriptional regulators have been characterized in actinomycetes and regulate a wide variety of physiological processes, including the specific activation of secondary metabolite biosynthesis [13]. In previous work, we constructed a *gsu1771*-deficient mutant strain using a markerless method (Δ*gsu1771* strain). The resulting Δ*gsu1771* strain exhibited higher rates of soluble and insoluble Fe(III) reduction than the wild-type strain and overexpresses *pilA* and the *c*-type cytochromes *omcB*, *omcE*, *omcS*, and *omcZ*. Additionally, the Δ*gsu1771* strain produces a thicker biofilm with higher exopolysaccharide production than the wild-type strain. Electrochemical characterization demonstrated that the Δ*gsu1771* biofilm grown on a fluorine-doped tin oxide (FTO) electrode exhibited a higher current output than that of the wild-type strain, indicating that GSU1771 controls genes related to extracellular electron transfer and electroconductive biofilm formation [14].

The aim of this study was to analyze the expression of genes in two different conditions of biofilm formation and with GSU1771-deficient mutant, which will allow to elucidate the signals that *G. sulfurreducens* detects to control the expression of certain cytochromes and genes involved in extracellular electron transfer. For this, we first characterized the GSU1771 regulon through RNA-seq analysis of biofilms grown on two different support materials: (1) a nonconductive material (glass, acetate-fumarate respiration) and (2) a conductive material in current production mode (graphite, acetate-electrode respiration). Transcriptome analysis of the Δ*gsu1771* biofilm grown on glass elucidated 467 differentially expressed (DE) genes with

respect to the wild-type strain, of which 167 were upregulated and 300 downregulated. The functions of these DE genes included energy metabolism and electron transport, transmembrane transport, exopolysaccharide production, signal transduction, and regulation, among others. In contrast, the RNA-seq analysis of the Δ*gsu1771* biofilm grown on graphite revealed 119 DE genes, several of which encoded proteins involved in the type VI secretion system, *c*-type cytochromes, and transport proteins. Furthermore, 56 of the DE genes identified in the biofilm grown on graphite were also identified in the biofilm grown on glass. RT-qPCR analyses were then conducted to confirm the differential expression of a selection of DE genes of interest including *pilA*, *pgcA*, *omcM*, *ppcD*, *csrA*, and *gsu3356* in the Δ*gsu1771* strain biofilm. Electrophoretic mobility shift assays were also conducted to characterize the interaction (i.e., binding) between the GSU1771 protein and different promoter regions of the *pgcA*, *pulF*, *gsu1771*, *gsu3356*, and *relA* genes. Overall, our findings demonstrated the key role of the GSU1771 regulator in controlling genes directly involved in the extracellular transfer of electrons, stress responses, and the formation of electroconductive biofilms.

## Materials and methods

### Bacterial strains and culture conditions

The bacterial strains, plasmids, and oligonucleotides used in the present study are summarized in S1 Table. The *G. sulfurreducens* strains were routinely grown in NBAF anoxic medium (acetate-fumarate) [15] at 30°C and *Escherichia coli* strains were grown in LB medium with ampicillin (200 μg/ml) at 37°C.

Total RNA was isolated from cells derived from biofilms grown on glass and graphite. For the first condition, the cells were cultured in NBAF medium at 25°C for 48 h and the biofilms generated at the bottom of the culture flask (glass) were washed and separated from the planktonic cells. For the second condition, biofilms grown on a graphite anode assembled in an H-type MCF with FWAF medium were washed with PBS buffer and collected for RNA extraction. The biofilms were then isolated and resuspended in 1 ml of fresh NBAF medium and 100 μl of RNA *later* (Invitrogen). The mixture was incubated on ice for 30 min and the cells were collected by centrifugation at 14,000 rpm for 2 min. The cell pellets were then stored at –70°C until required.

### DNA extraction and manipulations

Genomic DNA, plasmids, and PCR products were purified using the DNeasy blood and tissue kit (Roche), the High Pure Plasmid Isolation kit (Roche), and the GeneJET PCR purification kit (Thermo Scientific), respectively. *E. coli* transformations and other routine DNA manipulations were conducted following standard procedures [16].

### Analysis of biofilm production and structure by CLSM

The biofilm structure and the ratio of live cells to dead cells were determined by confocal laser scanning microscopy (CLSM). Glass and graphite plates were used as supports for biofilm formation inside hermetically sealed test tubes in anaerobic conditions with NBAF medium. Incubation was performed without shaking at 25°C for 48 h. All of the solutions used in the following procedures were sterile and anaerobic. After removing the electrodes from the culture medium, the planktonic cells were removed from the biofilm with a mixture of 0.002 M cysteine and 0.9% saline isotonic solution. Afterward, a mixture of dyes from the LIVE/DEAD® BacLight Bacterial Viability kit (0.00334 mM SYTO9 and 0.02 M propidium iodide) dissolved in 0.9% saline isotonic solution and 0.1 M cysteine was added to the samples. The

samples were dyed for 10 minutes, during which they were protected from any extraneous light sources. The dye was then washed with 0.002 M cysteine and 0.9% saline solution. Finally, images were captured with an Olympus FV1000 microscope at excitation wavelengths of 488 nm (green channel) and 559 nm (red channel). Imaging was performed using an immersion objective (LUMFLN 60 X 1.1 W). Fluorescence was obtained with a spectral detector at a 500–545 nm range (SDM560) for the green channel and a 570–670 nm range (Mirror) for the red channel. Images were acquired through the Z-axis of the biofilm at regular thickness intervals. Image analysis was performed using the Comstat2 (version 2.1) and Fiji (version 2.9.0) software [17,18].

## RNA extraction

RNA was extracted from cells recovered from the biofilms after 48 h of incubation on glass and two weeks of growth on the graphite electrode using the RNeasy mini kit (Qiagen) according to the manufacturer's instructions. Residual genomic DNA was digested via DNase I treatment (Thermo Scientific). The concentration and purity of the RNA samples were quantified using a NanoDrop 200c spectrophotometer (Thermo Scientific), after which sample integrity and quality were assessed with an Agilent 2100 Bioanalyzer. The extracted RNA was used for both RNA-seq and RT-qPCR analyses. The experiments were performed in duplicate, independent experiments.

## RNA-seq and data analysis

RNA-seq analyses were conducted using RNA samples extracted from *G. sulfurreducens* biofilms [strains DL1 (wild-type) and Δ*gsu1771*]. Illumina sequencing was performed at the Unidad Universitaria de Secuenciación Masiva y Bioinformática (UUSMB; National Autonomous University of Mexico, Mexico). All RNA samples were processed as previously described [19,20]. Briefly, ribosomal RNA was removed using the Ribominus kit (Thermo Scientific) and cDNA libraries were constructed using the TruSeq Stranded mRNA kit (Illumina), after which they were purified using the Zymoclean Gel DNA Recovery Kit (Zymo Research). Finally, the libraries were sequenced on an Illumina NextSeq 500 sequencer and differential expression analysis was performed on the IDEAMEX web server (http://www.uusmb.unam.mx/ideamex/) [21] using the 'edgeR', 'DESeq2', 'limma–voom', and 'NOISeq' packages. Differentially expressed (DE) genes were defined as those having a *p*-value <0.01 and a Log2 fold change >1.5, and candidate genes were exclusively selected among the genes that were differentially expressed according to all four of the aforementioned analysis methods. Functional annotation of the DE genes was conducted using the Kyoto Encyclopedia of Genes and Genomes (KEGG) database [22] using a custom R script. RNA-seq transcriptome data were deposited in the NCBI Gene Expression Omnibus database under accession number GSE223184.

## GSU1771 protein purification

The pBAD/His-GSU1771 plasmid [14] was transformed into the *E. coli* MC1061 (S1 Table) strain and the expression of the 6his-GSU1771 protein was induced with 0.2% arabinose at 37˚C for 4 h. The protein was purified under non-denaturing conditions using Ni-NTA resin (Qiagen) at 4˚C according to the manufacturer's instructions. The obtained 6his-GSU1771 purified protein was concentrated using Ultra 0.5 mL centrifugal filters (Amicon) and the elution buffer was replaced with storage buffer (HEPES 40 nM, KCl 50 mM, MgCl 8 mM). Protein concentrations were determined via the Bradford assay (Bio-Rad). The integrity and molecular mass of 6his-GSU1771 (approximately 28.46 kDa) were confirmed via SDS-PAGE.

## DNA gel mobility shift assay

Fragments of the regulatory regions of the *pgcA* (391-bp), *pulF* (191-bp), *gsu1771* (195-bp), *gsu3356* (200-bp), *omcM* (369-bp), and *omcB* (210-bp) genes were amplified through PCR using *G. sulfurreducens* genomic DNA and the corresponding oligonucleotide pairs listed in S1 Table. A 146-bp fragment corresponding to the intergenic region between *gsu1704* and *gsu1705* was used as a negative control. The PCR products were purified using the GeneJet PCR purification kit (Thermo Scientific). Binding assays were performed by mixing 100 ng of each PCR fragment and 100 ng of the control fragment at increasing concentrations (0.1, 0.25, 0.5, and 1.0 μM) of purified GSU1771 protein. The DNA-protein mix was incubated in 20 μl of binding buffer (40 mM HEPES, 8 mM MgCl$_2$, 50 mM KCl, 1 mM DTT, 0.05% NP-40, and 0.1 mg/ml BSA) at 30°C for 30 min [20]. Afterward, the reactions were separated on a 6% poly-acrylamide gel under native conditions in 0.5X TBE buffer. The gels were stained with ethidium bromide and visualized in a Gel Doc DZ imaging system (Bio-Rad).

## RT-qPCR

The expression of a selection of genes that were differentially expressed in the RNA-seq analysis was quantified by RT-qPCR. Total RNA from *G. sulfurreducens* biofilms was obtained as described above. cDNA was synthesized using the Revert Aid First Strand DNA Synthesis kit (Thermo Scientific) and the specific reverse oligonucleotides listed in S1 Table. RT-qPCR was then performed using the Maxima SYBR Green/ROX qPCR Master Mix (Thermo Scientific) on a Rotor-Gene® Q MDx instrument (Qiagen). The relative expression of the target genes was calculated with the Rotor-Gene Q Series Software using the $2^{-\Delta\Delta CT}$ method. The expression of the *gsu2822* gene was used as an internal control. All reactions were performed in triplicate and their average values were calculated.

## Cytochrome *c* content and immunoblot analysis

Cell-free protein extracts from *G. sulfurreducens* biofilms were prepared as previously described [14]. Biofilm cells were resuspended in 200 μl of B-PER II Bacterial Protein Extraction Reagent (Thermo Scientific) and incubated for 15 minutes at room temperature. Cell debris and non-lysed cells were removed by centrifugation at 14,000 rpm for 5 min. Total protein content was quantified via the Bradford assay (Bio-Rad). Afterward, 30 μg of proteins were separated via 15% SDS-PAGE and heme groups were stained with 3,3′,5,5′-tetramethylbenzidine following standard procedures [23,24]. The same concentration of protein used in heme-staining was separated as a control-loading protein and observed by coomassie staining (S1 Fig). The SDS-PAGE gels were visualized using a Gel Doc DZ imager (Bio-Rad).

Next, immuno-detection of the PilA, OmcS, and OmcZ proteins was conducted using 1, 10, and 100 μg of protein extract from *G. sulfurreducens* biofilms, respectively. The proteins were separated via 15% SDS-PAGE and transferred to nitrocellulose membranes (Merck-Millipore). Western blot analyses were then conducted using PilA-, OmcS-, and OmcZ-specific rabbit polyclonal antibodies [25–27]. The membranes were blocked with 10% low-fat dry milk overnight at 4°C, after which they were thoroughly washed with PBST (PBS, 0.3% Tween) and PBS. The primary anti-PilA 1:1000, anti-OmcS 1:1000, and anti-OmcZ 1:500 antibodies in PBS-BSA 0.3% were then added and incubated overnight with gentle agitation at 4°C. The membranes were then washed once more and subsequently treated with alkaline phosphatase-coupled anti-rabbit IgG secondary antibodies (Invitrogen) at a 1:5000 dilution in PBS-BSA 0.3%, after which they were allowed to incubate overnight with gentle shaking at 4°C. Finally, the membranes were revealed with 1 ml BCIP/NBT solution (Sigma-Aldrich).

The protein amount used for each western blot was observed by coomassie staining as a loading control (S1 Fig).

## Current production

The current production of each strain was compared in a two-chambered H-cell system with a continuous flow of acetate-containing medium (10 mM) as the electron donor and graphite stick anodes (65 cm$^2$) poised at 60 mV versus Ag/AgCl as the electron acceptor. Once current production was initiated, the anode chamber received a steady input of fresh medium as described previously [7].

## Results and discussion

### Biofilm CLSM analysis on two different supports

Previous studies have demonstrated that the Δ*gsu1771* mutant strain forms a thicker and more structured biofilm on FTO electrodes compared with the wild-type biofilm strain [14]. To investigate whether the Δ*gsu1771* strain also produces thick and structured biofilms on different support materials, biofilms were grown on both non-conductive (glass) and conductive (graphite electrode) support materials, after which their structures and key parameters were characterized through CLSM coupled with the Comstat2 and Fiji software [17,18]. Fig 1 shows the CLSM images of the three-dimensional structures of the biofilms produced by the DL1 wild-type and Δ*gsu1771* strains at 48 h on the glass and graphite surfaces, respectively. The wild-type strain formed a thinner and more continuous biofilm on the glass support than on the graphite surface, where it formed smooth aggregates. In contrast, the Δ*gsu1771* strain formed a thicker and more structured biofilm on both support materials compared to the wild-type strain. Particularly, the biofilm produced by the mutant strain exhibited column-like structures with channels, as seen in the top-down view in Fig 1, which were not present in the wild-type strain. This morphology was consistent with that of a biofilm grown in FTO in a recent study conducted by our work group [14]. On the graphite support, the Δ*gsu1771* mutant strain exhibited similar growth patterns to those observed on glass. Column-like structures and channels were also observed, and the biofilm formed by the mutant strain grown on graphite was thinner than the biofilm formed on the glass surface (Table 1).

Table 1 shows the biofilm parameters measured through image analysis using the Comstat2 and Fiji software. The thickness of the wild-type biofilm on the graphite support was almost three times higher than that of the wild-type biofilm grown on the glass support. On graphite, the roughness coefficient was also higher than on glass, which was consistent with the observed morphology on the side view images. The roughness coefficient is a measure of the variability in the height of the biofilm [28]. High roughness values indicate higher heterogeneity in the biofilm surface. Therefore, given that graphite is not a completely smooth material, higher roughness values were expected in the biofilm samples grown on the graphite electrodes. Finally, the viability (i.e., live/death cell ratio) of the wild-type cells that formed biofilms on the glass surface was slightly higher than that of the cells grown on the graphite surface. As seen in the CLSM images, the Δ*gsu1771* biofilm was thicker than the wild-type biofilm, particularly on the glass supports, which is consistent with previous characterizations of this mutant [14]. The following sections provide evidence of the role of GSU1771 as a regulator of important genes involved in the formation of electroactive biofilms, which provides insights into the molecular mechanisms underlying the phenotypes observed through confocal microscopy.

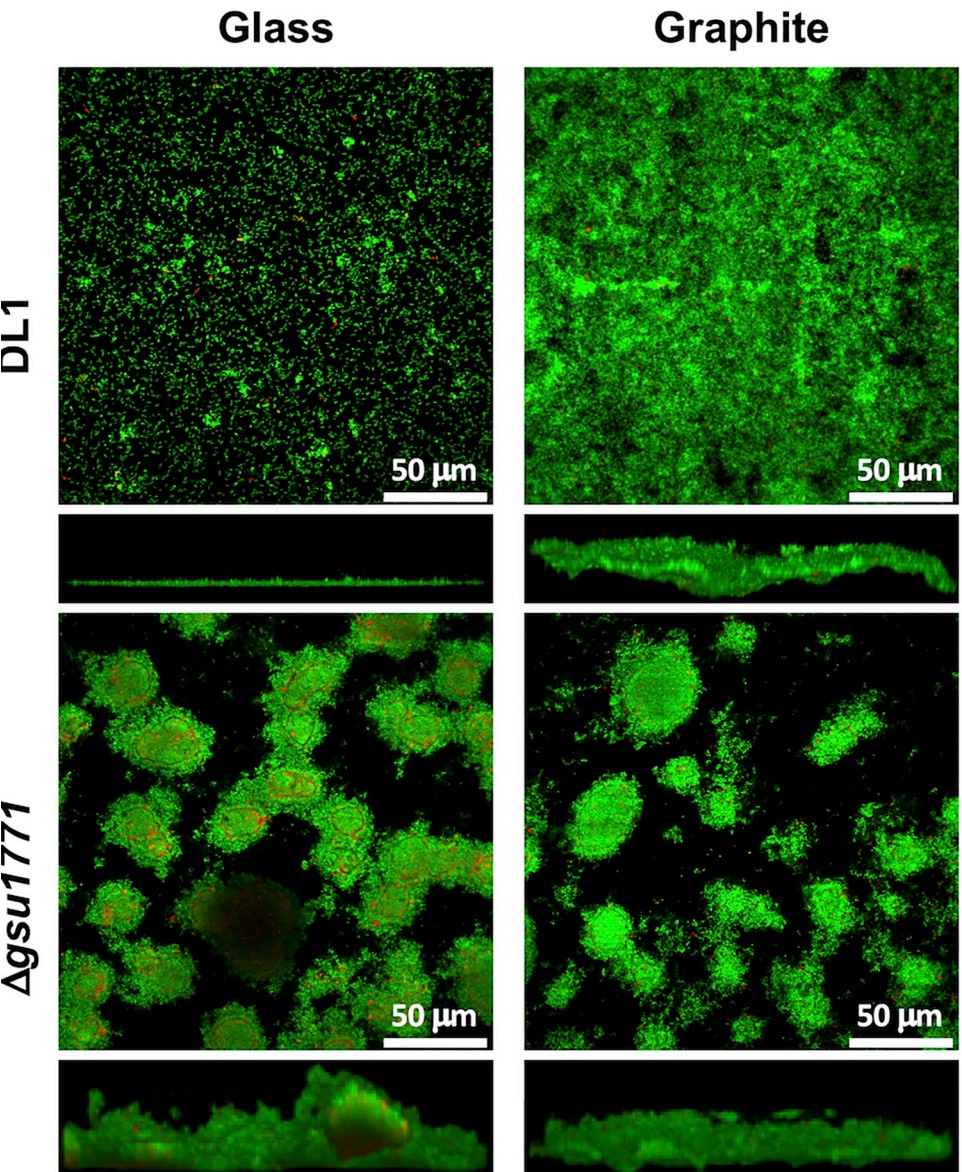

**Fig 1. CLSM images of wild-type (DL1) and Δ*gsu1771* biofilms formed on glass and graphite supports in fumarate-containing medium.** The top and bottom panels respectively illustrate the top and side view projections generated at 48 h of growth. Live and dead cells are indicated in green and red, respectively.

## Differentially expressed genes in biofilm grown on glass

To characterize the GSU1771 regulon in *G. sulfurreducens* during biofilm formation on a glass surface, the transcriptional profile of this bacterium was examined through RNA-seq analysis. DE genes were defined as those having a *p*-value <0.01 and a Log2FC >1.5 according to all four of the different analysis methods selected in the IDEAMEX platform (edgeR, DESeq2, limma–voom, and NOISeq). Based on these criteria, a total of 467 DE genes were identified (Fig 2A). After classifying the genes according to their LogFC values (positive or negative), 167 genes were upregulated and 300 were downregulated (Fig 2B) (S2 Table). The DE genes were then classified into the following functional categories according to KEGG enrichment analysis: "energy metabolism and electron transport," "carbohydrate metabolism," "transport,"

**Table 1. Biofilm parameters quantified from CLSM image analysis.**

| | Strain | | | |
|---|---|---|---|---|
| | DL1 (48 h of growth) | | Δ*gsu1771* (48 h of growth) | |
| | Glass | Graphite | Glass | Graphite |
| Thickness (μm) | 13.2 ± 0.8 | 39.3 ± 2.2 | 58.8 ± 6.2 | 39.3 ± 6.4 |
| Roughness coefficient | 1.77 ± 0.07 | 1.86 ± 0.03 | 1.23 ± 0.06 | 1.98 ± 0.01 |
| Cell viability (%) | 97.3 ± 0.7 | 93.7 ± 0.4 | 79.5 ± 1.4 | 92.1 ± 0.7 |

All the parameter values correspond to the average of n > 2 samples and their corresponding standard error (±).

"regulatory functions and transcription," "signal transduction," "cell envelope," "lipids metabolism," "nucleotide metabolism," "DNA/RNA metabolism," "protein synthesis," "proteolysis," "metabolism of proteins and cofactors," "unknown function," and "others" (Fig 2B).

"Energy metabolism and electron transport" was the category with the most DE genes (11.9%), which was followed closely by "regulatory functions and transcription" (10.4%),

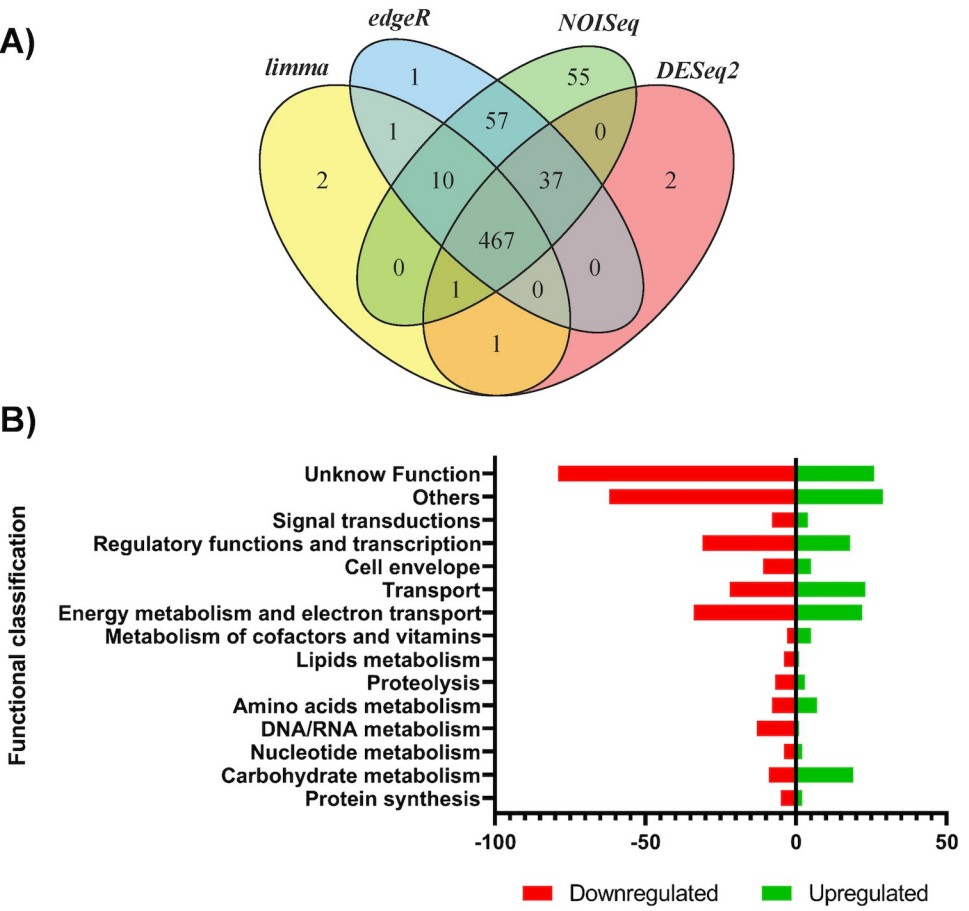

**Fig 2. Differential gene expression in Δ*gsu1771* biofilm versus wild-type biofilm (DL1) grown in glass support.** A) Venn diagram of DE genes identified using four different analysis methods. B) Functional overview of genes that were differentially expressed in the Δ*gsu1771* biofilm.

"transport" (9.6%), and "carbohydrate metabolism" (6%). Importantly, some of the DE genes in these functional categories are known to play important roles in biofilm formation and EET [7,9,10,29,30].

## Gene expression confirmation of selected genes with RT-qPCR

To validate the results obtained by the RNA-seq approach from biofilms grown in glass, RT-qPCR analyses were conducted on a selection of target genes. The selected genes encoded proteins involved in carbohydrate metabolism (*gsu1979* and *acnA*), energy metabolism and electron transport (*hybA*, *pgcA*, *omcM*, *ppcD*, and *pilA*), regulatory functions (*gnfK*, *gsu2507*, and *csrA*), transport (*dcuB* and *gsu0972*), lipid metabolism (*gsu0490*), amino acid metabolism (*gsu3142*), metabolism of cofactors and vitamins (*gsu1706*), cell envelope (*gsu0810*), and signal transduction (*gsu3356*) (Table 2). As expected, our RT-qPCR analyses confirmed the upregulation of *gsu1979*, *gsu0972*, *hybA*, *pgcA*, *omcM*, *gsu3142*, *gsu1706*, *gsu0941*, and *gsu2507*, as well as the downregulation of *dcuB*, *pilA*, *gsu0490*, *gsu0810*, *acnA*, *ppcD*, *csrA*, and *gsu3356* observed in Δ*gsu1771* during biofilm formation according to our RNA-seq analyses. Moreover, although the RNA-seq and RT-qPCR data exhibited differences in expression levels, both datasets showed the same regulation trends.

## Expression of *c*-type cytochromes and PilA

Given their critical role in direct EET, *c*-type cytochromes had been extensively studied in *G. sulfurreducens* [3]. According to our transcriptome analyses, 14 genes encoding *c*-type

**Table 2. RT-qPCR validation of differentially expressed genes elucidated by RNA-seq.**

| Locus ID | Name | Functional Annotation | Definition | Average n-fold change | Avg Δ*gsu1771*/Avg DL1 |
|---|---|---|---|---|---|
| *gsu1979* | *epsH* | Carbohydrate metabolism | exopolysaccharide synthesis membrane protein H (exosortase) | 2.116 | 98.87 |
| *gsu0972* | - | Transport | ATPase, AAA family / ABC transporter signature motif | 2.196 | 10.19 |
| *gsu0783* | *hybA* | Energy metabolism and electron transport | periplasmically oriented, membrane-bound [NiFe]-hydrogenase iron-sulfur cluster-binding subunit | 2.256 | 27.58 |
| *gsu1761* | *pgcA* | Energy metabolism and electron transport | lipoprotein cytochrome c | 2.029 | 22.5 |
| *gsu2294* | *omcM* | Energy metabolism and electron transport | Outer membrane cytochrome c | 5.309 | 67.18 |
| *gsu3142* | *aroG* | Amino acids metabolism | 3-deoxy-D-arabino-heptulosonate 7-phosphate synthase | 3.227 | 36.5 |
| *gsu1706* | *panC* | Metabolism of cofactors and vitamins | pantoate—beta-alanine ligase | 2.134 | 13.9 |
| *gsu0941* | *gnfK* | Regulatory functions and transcription | sensor histidine kinase | 3.151 | 20.81 |
| *gsu2507* | - | Regulatory functions and transcription | sensor histidine kinase, Cache_1 and HAMP domain-containing | 2.054 | 5.97 |
| *gsu0846* | *acnA* | Carbohydrate metabolism | aconitate hydratase 1 | -2.019 | 0.96 |
| *gsu0490* | *ato-I* | Lipid metabolism | succinyl:acetate coenzyme A transferase / Acyl-CoA hydrolase | -2.423 | 0.89 |
| *gsu0810* | - | Cell envelope | Proteína de membrana externa con Dominio OMP_b-brl y OmpA | -2.922 | 0.5 |
| *gsu2751* | *dcuB* | Transport | anaerobic C4-dicarboxylate antiporter, Dcu family | -2.606 | 0.69 |
| *gsu1496* | *pilA* | Energy metabolism and electron transport | geopilin domain 1 protein | -2.137 | 0.46 |
| *gsu1024* | *ppcD* | Energy metabolism and electron transport | cytochrome c | -3.095 | 0.3 |
| *gsu3041* | *csrA* | Regulatory functions and transcription | RNA-binding protein CsrA | -3.696 | 0.43 |
| *gsu3356* | - | Signal transduction | diguanylate cyclase, HAMP domain-containing | -2.074 | 0.74 |

cytochromes involved in energy metabolism and electron transport were differentially expressed (6 upregulated and 8 downregulated) (S2 Table).

The *c*-type cytochromes upregulated in the Δ*gsu1771* strain included *omcM*, *gsu2808*, *pgcA*, *gsu2495*, *gsu3615*, and *gsu2937*. Among these, *omcM* and *gsu2808* are also expressed during Fe (III) and Pd(II) reduction [3], whereas *gsu2937* also is expressed during Pd(II) reduction [19]. Additionally, *pgcA* codifies an extracellular cytochrome necessary for the reduction of Fe(III) and Mn(IV) oxides [31]. A recent study reported the potential involvement of PgcA in periplasmic electron transfer [32]. Furthermore, *gsu2495* encodes a periplasmic cytochrome that is upregulated in an *omcB*-deficient strain in the presence of Fe(III) oxide, whereas *gsu3615* is upregulated in the presence of acetate as an electron donor [33]. Finally, *gsu2937* encodes a periplasmic *c*-type cytochrome, which is upregulated during Pd(II) reduction and is possibly involved in selenite and tellurite reduction [19,34].

The *c*-type cytochromes downregulated in the Δ*gsu1771* strain were *macA*, *ppcD*, *gsu0068*, *gsu2811*, *gsu2743*, *gsu1740*, *gsu3259*, and *gsu2724*. MacA is an inner membrane cytochrome that participates in the reduction of Fe(III) and U(VI) oxides and forms a complex with PpcA [35–37]. PpcD is a tri-heme cytochrome involved in Fe(III) reduction and its expression increases during fumarate reduction using a graphite electrode as an electron donor [3,35,38]. Additionally, genetic studies have shown that *gsu0068* is important for insoluble Fe(III) reduction [3]. On the other hand, *gsu2811* is abundant in co-cultures of *Syntrophobacter fumaroxidans* and *G. sulfurreducens* compared to pure *G. sulfurreducens* cultures, as well as when the bacteria are grown in the presence of hydrogen as an electron donor [33,39]. The GSU2743 and GSU1740 cytochromes are abundantly expressed when formate is used as an electron donor and their expression is upregulated during fumarate reduction [33,40]. Moreover, the expression of *gsu2743* decreases in outer biofilms versus inner biofilms grown on a graphite anode that produces current [41]. GSU3259 (*imcH*) is an inner membrane cytochrome required for the reduction of electron acceptors with reduction potentials above −100 mV. Mutations in *imcH* have been reported to inhibit the reduction of Fe(III) citrate and Mn(IV) oxides [42]. GSU2724 is upregulated in the presence of formate and hydrogen as donor electrons [33]. Moreover, the transcript abundance of *gsu2724* increases during growth on Fe(III) and Mn(IV) oxide compared with growth on Fe(III) citrate, and the deletion of this gene impairs growth on Fe(III) oxide [3].

In a recent study, we reported increases in the total content of *c*-type cytochromes in planktonic cells of the Δ*gsu1771* strain grown in NBAF medium [14]. To determine whether this increase in *c*-type cytochromes was conserved in biofilms of the Δ*gsu1771* strain, the total content of *c*-type cytochromes was determined by SDS-PAGE and heme staining. As illustrated in Fig 3A, the whole-cell protein extracted from the Δ*gsu1771* biofilm exhibited an increase in the abundance of *c*-type cytochromes compared to the DL1 biofilm. Although the *omcS* and *omcZ* genes were not differentially expressed in the Δ*gsu1771* strain according to our RNA-seq analyses, western blot analyses were conducted to assess whether there were differences in their expression at the protein level. Our findings indicated that the OmcS and OmcZ cytochromes were upregulated in the Δ*gsu1771* biofilm compared to DL1. This included the active form of OmcZ (OmcZ$_S$), which is obtained through the post-translational processing of OmcZ$_L$ by OzpA, an enzyme that was also found to be upregulated in our RNA-seq analyses (Fig 3B and 3C, S2 Table) [43,44].

Surprisingly, the *pilA* gene was downregulated in the Δ*gsu1771* biofilm according to our RNA-seq analyses, which was contrary to its expression pattern during planktonic growth and Fe(III) reduction [12,14]. Therefore, immunodetection of the PilA protein was conducted to confirm whether its expression was correlated with that of the *pilA* transcript. As illustrated in Fig 3D, the content of PilA was similar in both the DLI and Δ*gsu1771* biofilms, suggesting the

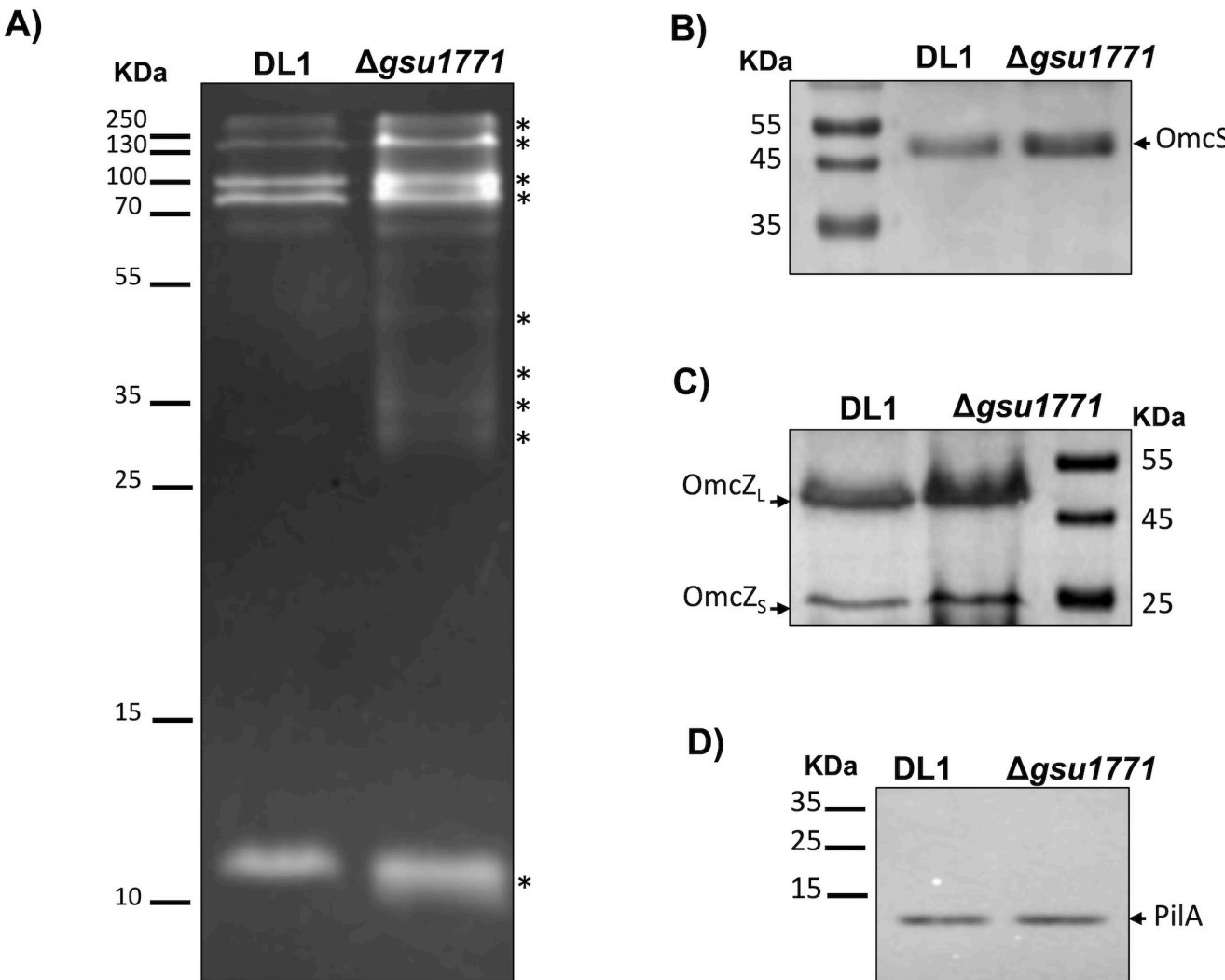

**Fig 3. Expression of *c*-type cytochromes and PilA content in Δ*gsu1771* and wild-type biofilms.** A) SDS-heme staining from whole biofilm extracts. The asterisks indicate the major *c*-type cytochromes in the Δ*gsu1771* biofilm. Western blot analysis of OmcS (B), OmcZ (C), and PilA (D) in biofilms. Heme-staining and western blot analysis are representatives of several replicates.

involvement of an unknown post-transcriptional regulatory mechanism that promotes the translation of PilA and other proteins such as OmcS and OmcZ in the Δ*gsu1771* biofilm.

## DE genes involved in exopolysaccharide production

Exopolysaccharides are essential components of the extracellular matrix of bacterial biofilms [45]. Previous studies in *G. sulfurreducens* have characterized the expression of the *xapD* (*gsu1501*) gene, which belongs to an ABC transporter-dependent exopolysaccharide production pathway, to investigate the role of exopolysaccharides in electroconductive biofilms. Exopolysaccharides promote the adherence between cells and abiotic surfaces, in addition to enabling the anchoring of *c*-type cytochromes in the extracellular matrix [9–11]. However, additional studies are needed to gain more insights into the role of the exopolysaccharides in the formation of electroconductive biofilms by *G. sulfurreducens*. RNA-seq analysis revealed a cluster of upregulated genes related to exopolysaccharide synthesis in the Δ*gsu1771* strain (S2

Table). Among this cluster of genes, *gsu1963* encodes a putative flippase, which is an essential protein in Wzx/Wzy-dependent exopolysaccharide synthesis pathways. The aforementioned gene cluster also included the *gsu1959*, *gsu1961-62*, *gsu1976-77*, *gsu1952*, and *gsu0991* genes, which encode putative glycosyltransferases that are essential for exopolysaccharide synthesis in bacteria [46]. The *gsu1985* and *epsH* genes were also identified within this cluster. Among them, *gsu1985* is a homolog to *epsE* of *Methylobacillus* sp. strain 12S, which encodes a polysaccharide co-polymerase enzyme (PCP) that participates in the synthesis of the polysaccharide methanolan in a Wzx/Wzy-dependent pathway. *epsH*, a member of the exosortase family of proteins, might also participate in methanolan synthesis. However, its specific role in this process remains unknown [47]. In addition to polysaccharide synthesis, EpsH might also be involved in protein export sorting [48].

Other genes related to exopolysaccharide synthesis that were upregulated in the Δ*gsu1771* strain included *neuB*, *gsu1972*, and *gsu1973*, which could be involved in the synthesis of sialic acid. These sugars have been mainly studied due to their role in concealing or masking pathogenic bacteria to avoid detection by the host's immune system [49]. Although the role of sialic acids in nonpathogenic bacteria is poorly understood, a recent study demonstrated that *neuB* mutation negatively affects biofilm formation, suggesting that sialic acid production could play an important role in the generation of biofilms by *G. sulfurreducens* [50].

The *gsu1958* and *gsu1980* genes, which encode putative polysaccharide deacetylase proteins, were upregulated in the Δ*gsu1771* strain. In *E. coli*, *Pseudomonas aeruginosa*, *Klebsiella pneumoniae*, and *Yersinia pestis*, deacetylation is a crucial step for polysaccharide maturation and correct positioning in the cell wall. However, in *G. sulfurreducens*, the role of acetylation in exopolysaccharide maturation remains unknown [51]. All of the above-described genes could thus belong to an uncharacterized pathway of exopolysaccharide synthesis in *G. sulfurreducens*, where exopolysaccharide modifications such as the addition of sialic acids and deacetylation are important contributors to electroactive biofilm formation [51].

## DE genes involved in transmembrane transport

Another functional category with a large number of differentially expressed genes in Δ*gsu1771* biofilm was "transport" (S2 Table). Among the upregulated genes in this category, the *pulGPQF* genes encode the proteins necessary to form a type II secretion system, which in turn is required for the secretion of proteins critical for Fe(III) and Mn(IV) reduction such as OmpB [52].

Additionally, we detected DE genes that encode subunits of the RND superfamily of transporters. RND transporters function as major drug efflux pumps in many gram-negative bacteria [53]. In the Δ*gsu1771* biofilm, the *gsu2135* and *gsu2136* genes were upregulated and encoded two of the three subunits of the RND-superfamily CzcABC transporter. In *G. sulfurreducens*, CzcABZ transporters are implicated in the export of metals from the periplasm across the outer membrane and play a relevant role in Co(II) detoxification [54]. In contrast, the *gsu1330-32* genes, which also encode an RND transporter, were downregulated in the Δ*gsu1771* biofilm [54].

ABC transporters are involved in the export and import of a wide range of molecules, such as the export of exopolysaccharides across the periplasm and through the outer membrane [55]. In the Δ*gsu1771* biofilm, we identified upregulation of the *lptFG* (*gsu1922-23*) genes, which are homologs to subunits of the ABC-transporter Lpt complex of *E. coli* [56]. Previous studies have also reported the upregulation of ABC-transporters in *G. sulfurreducens* MFCs, suggesting that the transport of extracellular matrix components plays an important role in electroactive biofilm formation [7,30]. Furthermore, the upregulation of genes that encode

transport systems putatively involved in the export of exopolysaccharides in the Δ*gsu1771* biofilm was consistent with the previously reported increase in the content of exopolysaccharides in this strain [14].

## Genes for regulation and signal transduction

Another functional category that contained a large number of DE genes in Δ*gsu1771* biofilm was "regulatory function and transcription" (S2 Table). Some of the genes in this functional category encoded two-component systems (TCSs). These TCSs consist of a histidine kinase (HK) that activates a response regulator (RR) by phosphotransfer, with the RR typically being a transcriptional regulator [57]. A total of 90 HKs and 93 RRs are encoded in the genome of *G. sulfurreducens*. Some of these RRs have been identified as enhancer-binding proteins (EBP). However, most of the TCSs in *G. sulfurreducens* have not been characterized [58]. In the present study, among the DE genes identified in Δ*gsu1771* biofilm via RNA-seq analysis, 12 were HKs (7 upregulated and 6 downregulated) and 9 were RRs (3 upregulated and 6 downregulated).

Among the upregulated genes identified in our study, *gsu0470* encodes a sigma-54 dependent RR that shares 45.41% of identity with Nla6 from *Myxococcus xanthus* DK1622. In *M. xanthus*, Nla6 is a key regulator of sporulation and the development of fruiting bodies [59]. Additionally, the *gsu0470-71* genes were upregulated in the Δ*gsu1771* biofilm. These genes encode a putative RR and HK respectively, which are homologs to the TCS ZraS/ZraR genes of *E. coli* and whose function is to activate the expression of genes in response to stress [60]. Among the downregulated TCS genes, *kdpD/kdpE* (*gsu2483* and *gsu2484*) are homologs to the KdpD/KpdE system of *E. coli* that regulates the *kdpFACB* operon in response to potassium limitation or salt stress [61]. The decrease in the transcription of the *kdpD/kdpE* system was correlated with the decrease in the transcription of the *kdpABC* genes that code for the putative potassium-transporting ATPase complex, suggesting that this regulatory mechanism was conserved in *G. sulfurreducens* (S2 Table).

Cyclic diguanylate (c-di-GMP) is an important second messenger in bacteria, which modulates many physiological processes including biofilm formation. This molecule is synthesized by enzymes containing the GGDEF domain known as diguanylate cyclases (DGCs) and is degraded by phosphodiesterase enzymes [62]. The genome of *G. sulfurreducens* harbors 29 genes that encode diguanylate cyclases. However, only a few have been characterized [5]. Among the DE genes detected in Δ*gsu1771* via RNA-seq analysis, 6 genes encode putative diguanylate cyclases (1 upregulated and 5 downregulated), whereas 2 were identified as phosphodiesterase-encoding genes (1 upregulated and 1 downregulated) (S2 Table). Furthermore, *gsu0895* was upregulated in the Δ*gsu1771* biofilm, whereas *gsu1037*, *gsu1937*, *gsu3356*, *gsu1400*, and *gsu1149* were downregulated. Additionally, the putative phosphodiesterase genes *gsu2622* and *gsu1007* were upregulated and downregulated, respectively. Nevertheless, additional studies are needed to characterize the role of these diguanylate cyclases and phosphodiesterases in Δ*gsu1771* biofilm formation.

In addition to transcriptional regulation, post-transcriptional regulation also contributes greatly to controlling global expression patterns. One of the most studied regulators that control post-transcriptional expression levels in bacteria is CsrA. In bacteria, CsrA binds to specific sequences in mRNA near or overlapping ribosome binding sites and the establishment of this mRNA-protein complex inhibits mRNA translation [63]. In our RNA-seq analyses, we identified a CsrA homolog (*gsu3041*) that was downregulated in the Δ*gsu1771* biofilm. In *E. coli* and related bacteria, CsrA plays a key role in controlling physiological processes such as carbon metabolism, virulence, motility, and biofilm development [63]. Although the role of

CsrA as a post-transcriptional regulator has not been studied in *G. sulfurreducens* and in phylogenetically related bacteria, its downregulation in the Δ*gsu1771* strain coupled with the increase in biofilm thickness in this strain suggests that this gene is involved in the regulation of biofilm development [14]. Furthermore, LepA is a highly conserved protein that participates in 30S ribosomal subunit biogenesis and translation initiation [64]. In this study, the *gsu1266* (*lepA*) gene was upregulated in the Δ*gsu1771* strain. Therefore, the upregulation of *lepA* and downregulation of *csrA* in Δ*gsu1771* could promote the increased translation of some genes involved in electroactive biofilm development.

Additionally, the *gsu2236* gene was upregulated in Δ*gsu1771*. This gene encodes a homolog to RelA, which is classified as a stringent factor. The RelA enzyme synthesizes hormone-like molecules such as (p)ppGpp (guanosine pentaphosphate) in response to nutrient starvation, particularly amino acid shortages [65]. The deletion of the *gsu2236* (*rel*$_{Gsu}$) gene in *G. sulfurreducens* causes a deficiency in Fe(III) reduction, suggesting that Rel$_{Gsu}$ regulates the expression of genes involved in Fe(III) reduction, in addition to participating in the response to various environmental stresses [65]. In strain Δ*gsu1771*, Rel$_{Gsu}$ upregulation could favor the expression of genes relevant to biofilm formation and EET.

## DE genes in biofilm grown on the surface of a graphite electrode

To compare the transcriptional response of the Δ*gsu1771* biofilm grown on glass versus graphite in current production mode, the global transcriptional response of Δ*gsu1771* biofilm grown on a graphite electrode in an MFC was analyzed through RNA-Seq analysis. The performance of the MFC is shown in Fig 4. The Δ*gsu1771* strain was more efficient in charge transfer, producing approximately 20% more charge than the wild-type strain, which was consistent with previously reported data [14]. Under these conditions, a total of 119 DE genes were identified after statistical analysis (79 upregulated and 40 downregulated). The DE genes were classified into the following metabolic categories: "unknown function," "energy metabolism and electron transport," "regulatory functions and transcription," "transport," "amino acid metabolism," "cell envelope," "proteolysis," "signal transduction," "carbohydrate metabolism," "lipids metabolism," "DNA/RNA metabolism," and "others" (Fig 5). The categories containing the most DE genes were "unknown function" (41), "energy and electron transport" (20), "regulatory functions and transcription" (17), and "transport" (13) (S3 Table).

Our RNA-seq analyses also revealed 67 DE genes that were shared between both Δ*gsu1771* biofilms grown in graphite and glass. Among them, 42 were upregulated, 25 were downregulated, and 56 genes shared the same type of regulation (Table 3). In contrast, 11 DE genes in the Δ*gsu1771* biofilm grown on the graphite electrode were counter-regulated with respect to the biofilm grown on the glass support (Table 3). Among these 11 genes that presented counter-regulation in both RNA-seq datasets (glass and graphite), 7 encode hypothetical proteins, 3 encode proteins related to transcriptional regulation, and *dcuB* encodes a fumarate transporter protein [66].

Among the DE genes upregulated in the Δ*gsu1771* strain in both conditions, the *omcM*, *gsu2808*, and *gsu3615* genes, which encode *c*-type cytochromes, were also identified. The upregulation of these cytochromes in the Δ*gsu1771* strain suggests that they play a relevant role in biofilm-related EET. Additionally, a gene cluster composed of *gsu0972-73*, *gsu0975-77*, *gsu0979*, *gsu0982-83*, *gsu0987-89*, and *gsu0991* encoding hypothetical proteins was upregulated in both conditions. Previous studies have proposed that the aforementioned genes can be acquired by lateral gene transfer; however, their function remains unknown [67]. The upregulated genes encoding proteins related to transcriptional regulation included *gsu0470*, *gsu0471*, *gsu1265*, *gsu1268*, and *gsu2670*.

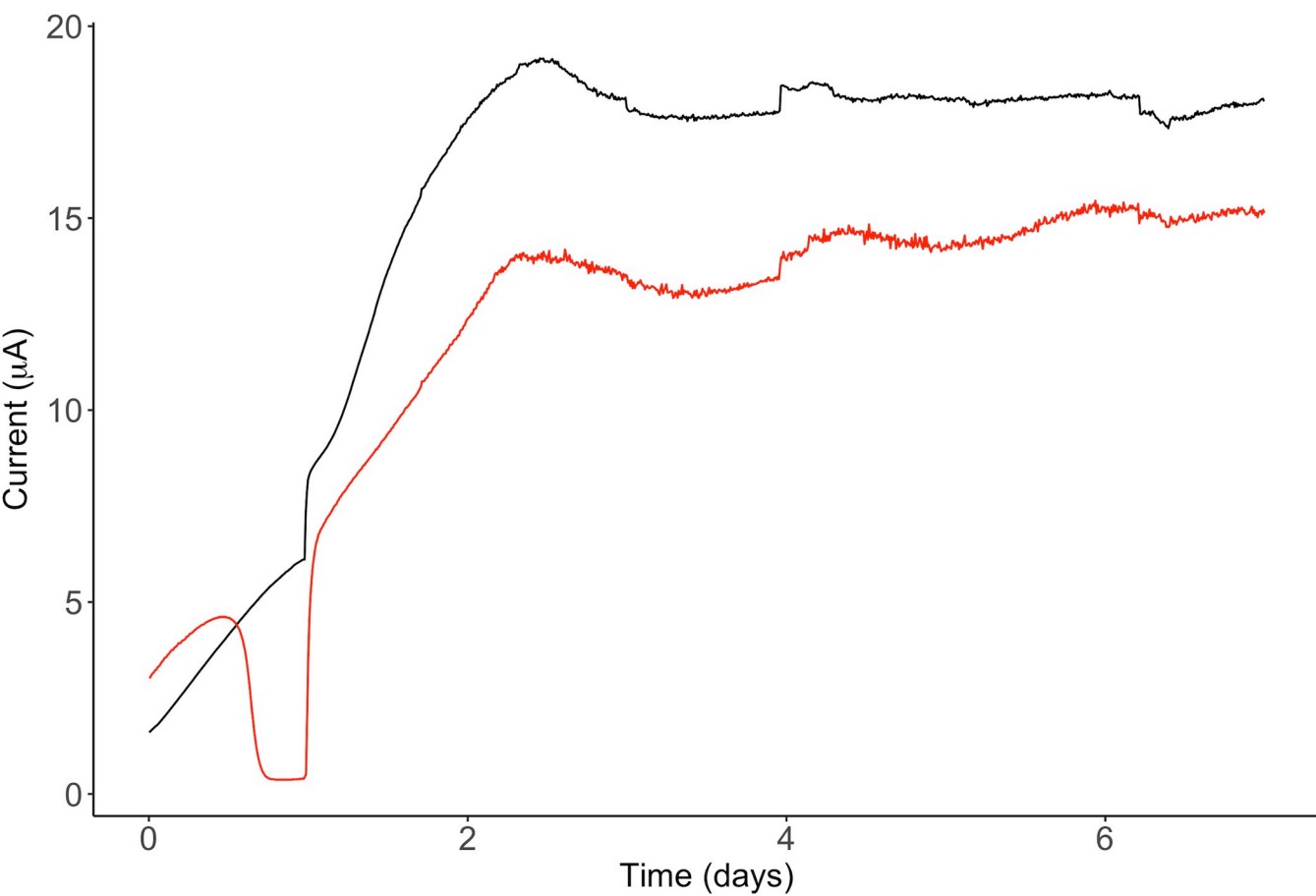

**Fig 4. Current production of *Geobacter sulfurreducens* strains.** The black and red lines represent the current production of the Δ*gsu1771* and wild-type strains, respectively, as a function of time. The data presented in the figure are representative time courses for multiple replicates for each treatment.

### DE genes detected exclusively in the graphite electrode biofilm

Our transcriptome analyses indicated that 52 genes (37 upregulated and 15 downregulated) were uniquely expressed in the Δ*gsu1771* biofilm formed on the graphite electrodes (Table 4). Importantly, the majority of these unique DE genes included *c*-type cytochromes (11), putative proteins of secretion systems (7), transcriptional regulators (7), and hypothetical proteins (10) (Table 4).

In the Δ*gsu1771* biofilm grown on the electrode, a group of genes encoding proteins homologous to the type VI secretion system (T6SS) were upregulated, including *gsu3167* (*vasJ*), *gsu0433* (*vasG*), *gsu0428* (*tssJ*), *gsu3166* (*ImpK*), *gsu3166* (*impL*), and *gsu3174* (*hcp*). In bacteria, T6SS has been linked to virulence, antibacterial activity, metal ion uptake, transport, and biofilm formation [68–71]. In *P. aeruginosa*, *P. fluorescens*, and *Acidovorax citrulli*, T6SS has been related to the formation of mature biofilms [70–72]. In *P. aeruginosa* and *A. citrulli*, mutations in the *hcp* gene had negative effects on the development of mature biofilms and exopolysaccharide production [71,72]. However, although *G. sulfurreducens* harbors the genes that comprise the T6SS, few efforts have been made to characterize their expression and function in this species [73]. To the best of our knowledge, our study is the first to demonstrate the potential participation of the T6SS in the formation of electroconductive mature biofilms in *G. sulfurreducens*.

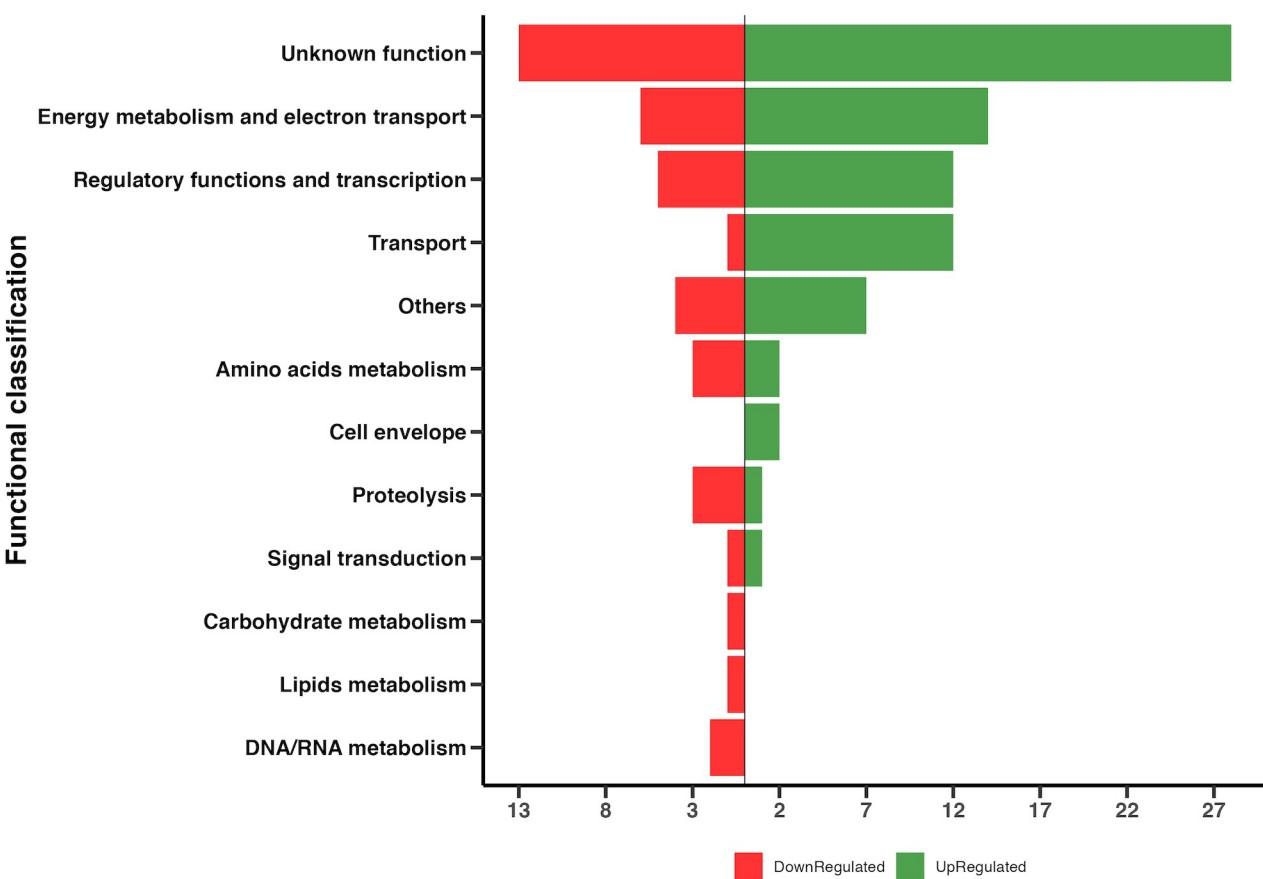

**Fig 5. Differential gene expression of the Δ*gsu1771* biofilm versus the wild-type biofilm (DL1) grown on the graphite electrode.** Functional overview of the DE genes detected in the Δ*gsu1771* biofilm.

The *c*-type cytochromes upregulated in the MFC included *omcQ*, *gsu1538*, *gsu2513*, *gsu2748*, *gsu2801*, *gsu2887*, *gsu0702*, and *gsu2642* (*omcW*) (Table 4). OmcQ is a 12-heme cytochrome that is also associated with Pd(II) reduction [19]. The *gsu1538* and *gsu2513* genes, which encode cytochromes, were upregulated in response to Co(II) accumulation in the periplasm [54]. Additionally, GSU1538 has a putative domain of cytochrome *c* peroxidase and is involved in the reduction of hydrogen peroxide to avoid oxidative stress damage. In bacteria, the activity of cytochrome *c* peroxidase is dependent on the availability of electrons from small mono-heme cytochromes, suggesting that GSU2513 is a redox partner of GSU1538 [54,74]. Therefore, the upregulation of the *gsu1538* and *gsu2513* genes in Δ*gsu1771* biofilm suggests their protective role against the oxidative stress generated from metabolism in MFCs. Moreover, previous studies reported that *in G. sulfurreducens* the *gsu2748* cytochrome was downregulated in a *rel*$_{Gsu}$ mutant and during growth on Fe(III) [65].

The *gsu2801* gene encoding a cytochrome related to the U(VI) response was highly expressed. However, its function remains unknown [75]. In another work, the *gsu2887* gene was associated with Fe(III) citrate reduction because its encoded protein was upregulated in the *omcF*-mutant [76]. Additionally, the GSU2887 lipoprotein cytochrome *c* was highly expressed in a co-culture of *Desulfotomaculum reducens* MI-1 and *G. sulfurreducens* in Fe(III)-reducing conditions [77]. A recent study reported that the *gsu0702* gene was upregulated in biofilms grown in graphite electrodes poised to a –0.17 V potential in the presence of acetate

Table 3. Differentially expressed genes in Δ*gsu1771* biofilm in both transcriptome data sets.

| Regulation | Locus ID | Name | Glass (Log2FC) | MFC (Log2FC) |
|---|---|---|---|---|
| Upregulated | | | | |
| | GSU0470 | Sigma 54-dependent transcriptional regulator | 2.41 | 3.03 |
| | GSU0471 | Two-component system, sensor histidine kinase | 2.55 | 3.61 |
| | GSU0619 | Hypotetical protein | 2.23 | 2.1 |
| | GSU0782 | Hydrogenase small subunit, *hybS* | 2.29 | 3.42 |
| | GSU0973 | Hypotetical protein | 2.51 | 2.46 |
| | GSU0974 | Hypotetical protein | 2.12 | 2.56 |
| | GSU0975 | Phage tail sheath protein | 2.53 | 2.73 |
| | GSU0976 | Phage tail tube protein gp19 | 2.6 | 2.53 |
| | GSU0978 | Hypotetical protein | 2.57 | 2.13 |
| | GSU0982 | Phage protein D | 2.15 | 2.07 |
| | GSU0983 | Phage tail spike protein | 2.33 | 1.93 |
| | GSU0986 | Phage baseplate outer wedge protein | 2.09 | 2.24 |
| | GSU0987 | Hypotetical protein | 2.8 | 2.53 |
| | GSU0988 | Hypotetical protein | 2.52 | 2.3 |
| | GSU0989 | NHL repeat domain protein | 2.51 | 2.61 |
| | GSU0990 | Hypotetical protein | 3.18 | 3.5 |
| | GSU0991 | Glycosyltransferase | 2.94 | 3.19 |
| | GSU0992 | Hypotetical protein | 3.509 | 3.01 |
| | GSU1265 | Sensor histidine kinase response regulator | 3.87 | 4.07 |
| | GSU1268 | Helix-turn-helix transcriptional regulator, LysR family | 3.47 | 3.47 |
| | GSU1442 | Carbonic anhydrase | 2.35 | 3.98 |
| | GSU2294 | OmcM; cytochrome *c* | 5.3 | 3.97 |
| | GSU2670 | Helix-turn-helix transcriptional regulator, LuxR family | 1.98 | 3.56 |
| | GSU2808 | Lipoprotein, cytochrome *c* | 1.72 | 2.69 |
| | GSU3615 | Cytochrome *c* | 2.62 | 1.62 |
| | GSU3141 | Hypotetical protein | 2.67 | 1.84 |
| | GSU3142 | 3-deoxy-7-phosphoheptulonate synthase, *aroG-2* | 3.22 | 1.94 |
| | GSU0972 | ATPase, AAA family | 2.19 | 2.44 |
| | GSU0977 | Hypotetical protein | 2.44 | 2.7 |
| | GSU0979 | Phage tail tube protein gp19, putative | 2.1 | 1.73 |
| | GSU0980 | Hypotetical protein | 1.96 | 1.76 |
| | GSU1640 | Cytochrome bd ubiquinol oxidase subunit I, *cydA* | 2.25 | 2.1 |
| | GSU2410 | HSP20 family protein, *hspA-2* | 1.54 | 3.33 |
| | GSU2939 | Outer membrane channel | 1.58 | 1.62 |
| | GSU2940 | Rhodanese homology domain pair protein | 1.65 | 1.94 |
| Downregulated | | | | |
| | GSU2585 | Hypotetical protein | -5.24 | -1.69 |
| | GSU2614 | Single-stranded-DNA-specific exonuclease, *recJ* | -2.166 | -2.42 |
| | GSU0919 | Hypothetical protein | -1.65 | -2.12 |
| | GSU0071 | Hypotetical protein | -4.016 | -3.21 |
| | GSU0081 | Hypotetical protein | -1.76 | -1.72 |
| | GSU0216 | Hypotetical protein | -2.59 | -2.87 |
| | GSU0547 | DNA mismarch repair protein, *mutS-2* | -3.5 | -3.19 |
| | GSU0548 | Radical SAM domain | -4.67 | -3.44 |

(*Continued*)

**Table 3.** (Continued)

| Regulation | Locus ID | Name | Glass (Log2FC) | MFC (Log2FC) |
|---|---|---|---|---|
| | GSU1037 | Diguanylate cyclase/phosphodiesterase | -2.93 | -3.56 |
| | GSU1394 | laccase family multicopper oxidase, *ompB* | -1.54 | -1.94 |
| | GSU1395 | Hypotetical protein | -1.57 | -2.21 |
| | GSU1877 | Nitronate monooxygenase | -3.75 | -3.36 |
| | GSU1943 | PEP motif-containing protein, putative exosortase substrate | -2.87 | -2.72 |
| | GSU3568 | Pseudogene, *lnt-C* | -1.77 | -3.06 |
| | GSU1079 | PEP motif-containing protein, putative exosortase substrate | -2.25 | -2.23 |
| | GSU1513 | SAM-dependent methyltransferase | -2.45 | -2.95 |
| | GSU1944 | PEP motif-containing protein, putative exosortase substrate | -3.32 | -2.6 |
| | GSU2487 | Carbamate kinase, *cpkA* | -2.82 | -1.55 |
| | GSU2584 | Lipoprotein | -4.76 | -1.61 |
| | GSU2662 | Membrane protein | -2.65 | -4.2 |
| | GSU3329 | Radical SAM domain iron-sulfur cluster-binding oxidoreductase | -1.68 | -1.82 |
| Counter-regulated genes | | | | |
| | GSU2505 | Hypotetical protein | 1.94 | -4.97 |
| | GSU2506 | Sigma-54-dependent sensor transcriptional response regulator, PAS domain-containing | 1.82 | -2.22 |
| | GSU2499 | Hypothetical protein | 1.83 | -1.94 |
| | GSU2507 | Sensor histidine kinase | 2.05 | -1.59 |
| | GSU0597 | Hypotetical protein | -1.87 | 2.26 |
| | GSU3489 | Hypotetical protein | -1.91 | 1.89 |
| | GSU0596 | Response receiver | -1.81 | 2.06 |
| | GSU2750 | Hypothetical protein | -2.03 | 2.02 |
| | GSU2751 | Anaerobic C4-dicarboxylate transporter, *dcuB* | -2.6 | 1.92 |
| | GSU3409 | Hypotetical protein | -3.26 | 2.92 |
| | GSU3410 | Hypotetical protein | -3.37 | 2.06 |

with respect to biofilms poised with the same potential in the presence of formate [78]. Although this large GSU0702 cytochrome was predicted to be extracellular, its function outside the cell has not been established. Finally, *omcW* (*gsu2642*), which is predicted as an outer surface cytochrome, was upregulated in cells grown in Fe(III) oxides but not in citrate Fe(III). In contrast, a mutant strain lacking *omcW* exhibited no Fe(III) reduction phenotype [3,79].

Unlike in planktonic cells and glass-generated biofilms, *omcS* was downregulated in electrode-grown biofilms. *omcS* is transcribed in a transcriptional unit and in an operon together with *omcT* [80]. *omcS* and *omcT* are expressed in the presence of Fe(III) oxides and fumarate as electron acceptors. However, only *omcS* is necessary for the reduction of Fe(III) oxides. In the Δ*gsu1771* biofilm grown on the graphite electrodes, *omcT* was also downregulated. The change in the regulation of *omcS* expression in biofilms grown on MFCs suggests that other regulatory proteins could be involved under these conditions, resulting in different transcriptional control mechanisms than those previously reported.

**Table 4. Differentially expressed genes in Δ*gsu1771* biofilm only in graphite electrode.**

| Regulation | Locus ID | Name | Log2FC | FDR |
|---|---|---|---|---|
| Upregulated | GSU0537 | Sensor diguanylate cyclase/phosphodiesterase | 2.26 | 5.98E-05 |
| | GSU0538 | HSP20 family protein, *hspA-1* | 2.36 | 1.85E-05 |
| | GSU0592 | Cytochrome *c*, *omcQ* | 2.12 | 1.78E-07 |
| | GSU1154 | Surface repeat protein | 2.39 | 1.67E-07 |
| | GSU1264 | Histidine phosphotransfer domain protein | 7.60 | 3.86E-22 |
| | GSU1538 | Cytochrome *c* | 4.04 | 7.67E-11 |
| | GSU1556 | Lipoprotein | 2.24 | 4.13E-07 |
| | GSU1948 | Hypothetical protein | 2.10 | 3.97E-06 |
| | GSU3548 | Type IV pilus minor pilin PilE, *pilE* | 3.08 | 1.25E-08 |
| | GSU2172 | Peptidoglycan-binding protein | 4.37 | 6.35E-12 |
| | GSU2513 | Lipoprotein cytochrome *c* | 1.91 | 4.43E-06 |
| | GSU2748 | Cytochrome *c* | 2.16 | 2.23E-06 |
| | GSU2749 | NOL1/NOP2/Sun family protein | 2.40 | 6.22E-07 |
| | GSU2801 | Cytochrome *c* | 2.34 | 2.32E-09 |
| | GSU2887 | Cytochrome *c* | 1.67 | 7.88E-05 |
| | GSU2967 | Ferritin-like domain protein | 4.03 | 2.67E-13 |
| | GSU2968 | Hypothetical protein | 3.60 | 3.34E-09 |
| | GSU3171 | Hypothetical protein | 3.72 | 1.94E-09 |
| | GSU3261 | Response regulator, putative | 2.45 | 1.16E-07 |
| | GSU3419 | Sensor histidine kinase | 2.40 | 1.27E-07 |
| | GSUR0059 | 6S RNA, stationary phase repressor of sigma-70-containing RNA polymerase | 3.03 | 6.21E-08 |
| | GSU3167 | Type VI secretion system protein VasJ | 5.35 | 2.62E-15 |
| | GSU0433 | Type VI secretion system protein VasG | 2.41 | 7.96E-09 |
| | GSU0428 | Type VI secretion system outer membrane lipoprotein TssJ, *tssJ* | 7.88 | 6.19E-07 |
| | GSU0702 | Cytochrome *c* | 2.13 | 2.45E-07 |
| | GSU0981 | Hypothetical protein | 1.91 | 0.000138 |
| | GSU1018 | Hypothetical protein | 2.31 | 8.26E-09 |
| | GSU1153 | Outer membrane lipoprotein | 2.03 | 4.43E-06 |
| | GSU1905 | Cold shock protein | 1.65 | 0.000527 |
| | GSU1945 | Fibronectin type III domain protein | 2.10 | 0.000326 |
| | GSU1947 | Hypothetical protein | 1.60 | 0.001374 |
| | GSU2642 | Cytochrome *c* | 2.64 | 9.90E-05 |
| | GSU3165 | Type VI secretion system protein ImpK | 3.71 | 6.68E-07 |
| | GSU3166 | Ttype VI secretion system protein ImpL | 3.88 | 5.52E-15 |
| | GSU3174 | Type VI secretion system secreted protein Hcp | 4.13 | 2.50E-08 |
| | GSU3370 | Helix-turn-helix transcriptional regulator, GntR family | 1.78 | 3.97E-06 |
| | GSU0431 | Type VI secretion system protein ImpG | 4.38 | 2.47E-06 |
| Downregulated | GSU1238 | Iron-sulfur cluster-binding protein | -2.44 | 2.58E-07 |
| | GSU1512 | Hypothetical protein | -2.38 | 2.01E-08 |
| | GSU1514 | Heptosyltransferase family protein | -3.11 | 1.88E-07 |
| | GSU1939 | Sensor histidine kinase | -2.24 | 2.23E-07 |
| | GSU2502 | Spermidine synthase | -1.96 | 1.02E-06 |
| | GSU2503 | Cytochrome *c*, *omcT* | -5.03 | 2.81E-28 |
| | GSU2815 | Sensor histidine kinase | -2.49 | 2.53E-11 |
| | GSU3085 | Dimetal-binding protein, *yqfO* | -1.53 | 0.000124 |
| | GSU2504 | Cytochrome *c*, *omcS* | -4.99 | 4.71E-23 |
| | GSU0545 | Hypothetical protein | -1.75 | 2.31E-05 |

*(Continued)*

**Table 4.** (Continued)

| Regulation | Locus ID | Name | Log2FC | FDR |
|---|---|---|---|---|
| | GSU1237 | Pyridine nucleotide-disulfide oxidoreductase family protein | -2.00 | 6.61E-07 |
| | GSU3586 | YVTN family beta-propeller domain protein | -1.67 | 6.91E-05 |
| | GSU2501 | Cytochrome *c* | -1.69 | 4.55E-05 |
| | GSU2612 | Rubrerythrin/rubredoxin protein | -1.86 | 3.56E-06 |
| | GSU2613 | Cation efflux family protein, *fieF* | -1.92 | 8.65E-06 |

Four putative HKs (*gsu1264*, *gsu3419*, *gsu1939*, and *gsu2815*) and one RR (*gsu3261*) exhibited transcriptional changes in the Δ*gsu1771* electrode biofilm. Among these genes, *gsu1264*, *gsu3419*, and *gsu3261* were upregulated, whereas *gsu1939* and *gsu2815* were downregulated (Table 4). The function of the products of these genes is unknown. However, the *gsu3162* and *gsu2815* genes have been associated with the reduction of Fe(III) oxides and Pd(II) [12,24]. The *gsu3370* gene encodes a member of the GntR family regulators and was upregulated. In *G. sulfurreducens*, *gsu3370* is a transcriptional regulator that binds to the promoter region of *gltA*, an enzyme that plays a key role in the tricarboxylic acid cycle. Nevertheless, little is known regarding its regulatory mechanisms and its role in controlling central metabolism [81]. In bacteria, c-di-GMP is a major signaling molecule that modulates several bacterial functions such as virulence, motility, and biofilm formation [62]. The *gsu0537* gene encodes a putative diguanylate cyclase and was upregulated in the Δ*gsu1771* electrode biofilm. Previous studies have also reported that *gsu0537* was upregulated in an adapted *omcB*-mutant associated with Fe(III) and Pd(II) reduction [3,19]. Transcriptional changes were also identified in five genes containing riboswitch regulation elements of the GEMM-I (genes for environment, membranes, and motility) family, which respond to cyclic dinucleotides (Table 4). Moreover, *gsu1556*, *gsu1948*, *gsu1018*, and *gsu1945* were upregulated and *omcS* was downregulated. Nevertheless, the function of these positively regulated genes and the mechanism of the GSU1771 regulator remain uncharacterized.

## GSU1771 binding to the promoter regions of *pgcA*, *pulF*, *gsu1771*, *gsu3356*, and *relA*

To determine whether GSU1771 directly regulates the expression of the *pgcA*, *pulF*, *gsu1771*, *gsu3356*, and *relA* genes, we analyzed the binding of GSU1771 to their promoter regions through electrophoretic mobility shift assays (EMSA). The *gsu1704-gsu1705* intergenic region (control) and the *omcB* promoter region were used as negative controls [14]. As shown in Fig 6, GSU1771 was bound directly to all of the examined sequences except the negative control, meaning that this protein specifically interacted with all of the promoter regions. Taken together, our findings suggest that this regulator acts as a repressor of *pgcA*, *pulF*, and *relA* and as a transcriptional activator of *gsu1771* and *gsu3356*.

## Conclusions

Biofilm production is an important process in *G. sulfurreducens* due to its biotechnological applications in bioelectricity production and bioremediation. However, little is known regarding the physiological processes involved in biofilm production in *G. sulfurreducens*, as well as the regulatory mechanisms that govern them. Here, we characterized the transcriptional responses of Δ*gsu1771* biofilm grown on a glass surface (non-conductive) and on an MFC graphite electrode in current production mode. CLSM analyses demonstrated that the Δ*gsu1771* strain forms a ticker biofilm on both support materials. Moreover, transcriptomic

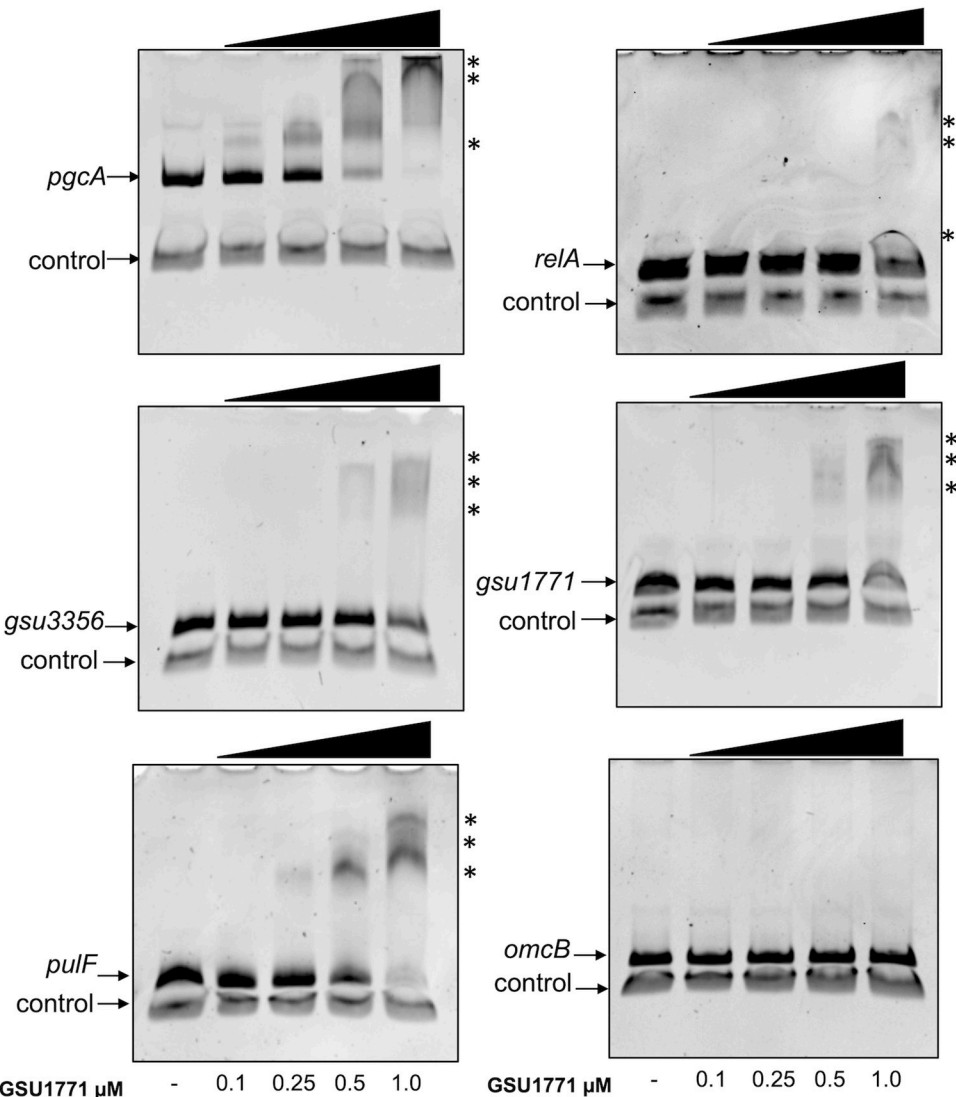

**Fig 6. GSU1771 protein binding to the DNA of promoter regions.** DNA-protein binding was assessed via competitive non-radioactive EMSA. DNA of the promoter region of *pgcA*, *gsu1771*, *pulF*, *relA*, *gsu3356*, and *omcB* was incubated with increasing concentrations of purified GSU1771 (0, 0.1, 0.25, 0.5, and 1 μM). A fragment containing the *gsu1704-gsu1705* intergenic region was included in each reaction as a negative control. The asterisks indicate the DNA-protein complexes.

analysis of the Δ*gsu1771* biofilm grown on the surface of glass and graphite electrodes revealed DE genes with respect to the wild-type strain in both conditions. The DE genes of the biofilms grown on glass belonged to different metabolic categories, including genes from a putative pathway of exopolysaccharide synthesis, as well as genes involved in transmembrane transport, energy metabolism, signal transduction, and transcriptional regulation. The Δ*gsu1771* biofilm grown on the MFC graphite electrode shared several DE genes with biofilm grown on glass, several of which encoded *c*-type cytochromes involved in EET, in addition to transcriptional regulators and a group of genes that are reportedly acquired by horizontal gene transfer. Additionally, the Δ*gsu1771* biofilm grown on the graphite electrode surface exhibited several interesting unique DE genes that encode *c*-type cytochromes. These genes could be targeted in future studies to enhance current production in MFCs, as well as to determine whether T6SS

genes contribute to biofilm maturation in *G. sulfurreducens*. Additionally, our study also identified genes that encode putative proteins involved in signal transduction, whose role in biofilm production and EET remains unknown. Finally, our EMSA results demonstrated that GSU1771 directly binds to the promoter region of several genes selected from our transcriptome analysis, thereby regulating their expression.

## Supporting information

**S1 Fig.** SDS-PAGE of protein used as a loading control in heme-staining (A) and western blot for OmcS (B), OmcZ (C) and PilA (D). The PageRuler Pre-stained Protein Ladder standard (ThermoScientific) was used as a molecular weight.
(DOCX)

**S1 Table. Bacteria, plasmid, and oligonucleotides used in this study.**
(DOCX)

**S2 Table. List of differentially expressed genes in Δ*gsu1771* compared with the DL1 strain during biofilm formation on glass.**
(DOCX)

**S3 Table. List of differentially expressed genes in Δ*gsu1771* compared with the DL1 strain during biofilm formation on graphite electrodes.**
(DOCX)

**S1 Raw images.**
(PDF)

## Acknowledgments

We thank Leticia Olvera, Veronica Jimenez-Jacinto, Ricardo Grande, Andres Saralegui, and Shirley Ainsworth for their technical support. We also thank Prof. Derek Lovley (University of Massachusetts) for kindly offering his laboratory space and equipment to perform MFC experiments. Trevor Woodard and Xinying Liu for advising on performing the MFC experiments and providing the anti-OmcS and anti-OmcZ antibodies used in this study. Oligonucleotides and automated sequencing were performed at the Unit for DNA Sequencing and Synthesis (IBT-UNAM).

## Author Contributions

**Conceptualization:** Juan B. Jaramillo-Rodríguez, Alberto Hernández-Eligio, Katy Juarez.

**Data curation:** Leticia Vega-Alvarado, Alberto Hernández-Eligio.

**Formal analysis:** Juan B. Jaramillo-Rodríguez, Leticia Vega-Alvarado, Luis M. Rodríguez-Torres, Guillermo A. Huerta-Miranda, Alberto Hernández-Eligio, Katy Juarez.

**Funding acquisition:** Katy Juarez.

**Investigation:** Juan B. Jaramillo-Rodríguez, Alberto Hernández-Eligio, Katy Juarez.

**Methodology:** Juan B. Jaramillo-Rodríguez, Luis M. Rodríguez-Torres, Guillermo A. Huerta-Miranda, Alberto Hernández-Eligio, Katy Juarez.

**Resources:** Katy Juarez.

**Software:** Leticia Vega-Alvarado.

**Supervision:** Alberto Hernández-Eligio.

**Validation:** Juan B. Jaramillo-Rodríguez, Alberto Hernández-Eligio, Katy Juarez.

**Visualization:** Juan B. Jaramillo-Rodríguez, Luis M. Rodríguez-Torres, Guillermo A. Huerta-Miranda.

**Writing – original draft:** Juan B. Jaramillo-Rodríguez, Guillermo A. Huerta-Miranda, Alberto Hernández-Eligio, Katy Juarez.

**Writing – review & editing:** Juan B. Jaramillo-Rodríguez, Alberto Hernández-Eligio, Katy Juarez.

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
