## [Decision Letter · Decision Letter 0]

7 Jun 2023

PONE-D-23-08621Global transcriptional analysis of Geobacter sulfurreducensgsu1771 mutant biofilm grown on two different support structuresPLOS ONE

Dear Dr. Juarez,

Thank you for submitting your manuscript to PLOS ONE. After careful consideration, we feel that it has merit but does not fully meet PLOS ONE’s publication criteria as it currently stands. Therefore, we invite you to submit a revised version of the manuscript that addresses the points raised during the review process.

We look forward to receiving your revised manuscript.

Kind regards,

Moupriya Nag

Academic Editor

PLOS ONE

Journal Requirements:

3. Please expand the acronym “PAPIIT-UNAM, CONACYT, PASPA-UNAM” (as indicated in your financial disclosure) so that it states the name of your funders in full.

"KJ and AH-E received  PAPIIT-UNAM (grant No IN212022).

BJ-R received CONACYT-scholarship awarded during his master’s program

GMH-M received CONACYT-postdoctoral fellowship (2322131)

KJ received PASPA-UNAM for financial support during a sabbatical stay"

"This study was financially supported by PAPIIT-UNAM (grant No IN212022). We thank Leticia Olvera, Veronica Jacinto, Ricardo Grande, Andres Saralegui, and Shirley Ainsworth for their technical support. BJ-R thanks CONACYT for the scholarship awarded during his master’s program. GMH-M thanks CONACYT for the postdoctoral fellowship (2322131), KJ also thanks PASPA-UNAM for financial support during a sabbatical stay.."

"KJ and AH-E received  PAPIIT-UNAM (grant No IN212022).

BJ-R received CONACYT-scholarship awarded during his master’s program

GMH-M received CONACYT-postdoctoral fellowship (2322131)

KJ received PASPA-UNAM for financial support during a sabbatical stay"

6. Thank you for stating the following in your Competing Interests section:  "NO authors have competing interests"

7. In your Data Availability statement, you have not specified where the minimal data set underlying the results described in your manuscript can be found. PLOS defines a study's minimal data set as the underlying data used to reach the conclusions drawn in the manuscript and any additional data required to replicate the reported study findings in their entirety. All PLOS journals require that the minimal data set be made fully available. For more information about our data policy, please see http://journals.plos.org/plosone/s/data-availability.

8. PLOS ONE now requires that authors provide the original uncropped and unadjusted images underlying all blot or gel results reported in a submission’s figures or Supporting Information files. This policy and the journal’s other requirements for blot/gel reporting and figure preparation are described in detail at https://journals.plos.org/plosone/s/figures#loc-blot-and-gel-reporting-requirements and https://journals.plos.org/plosone/s/figures#loc-preparing-figures-from-image-files. When you submit your revised manuscript, please ensure that your figures adhere fully to these guidelines and provide the original underlying images for all blot or gel data reported in your submission. See the following link for instructions on providing the original image data: https://journals.plos.org/plosone/s/figures#loc-original-images-for-blots-and-gels. 

9. Please amend either the title on the online submission form (via Edit Submission) or the title in the manuscript so that they are identical.

Reviewers' comments:

Reviewer's Responses to Questions

**Comments to the Author**

1. Is the manuscript technically sound, and do the data support the conclusions?

Reviewer #1: Yes

Reviewer #2: Partly

2. Has the statistical analysis been performed appropriately and rigorously? 

Reviewer #1: N/A

Reviewer #2: N/A

3. Have the authors made all data underlying the findings in their manuscript fully available?

Reviewer #1: Yes

Reviewer #2: No

4. Is the manuscript presented in an intelligible fashion and written in standard English?

Reviewer #1: Yes

Reviewer #2: Yes

5. Review Comments to the Author

Reviewer #1: The no of references should be reduced.

The manuscript contains voluminous data , that often confuses the reader as the objectives of the work is not mentioned clearly in the Introduction.

However the work is good and deserves publication after minor revision.

Reviewer #2: Appears to be interesting paper following up on prior observations. The entire paper is based on transcriptional analysis but reviewer is unable to review without access to RNAseq data. NCBI DEO database data locked until 2025 without secure token.

No data on depth, % that was rRNA, or quality of RNA read data.

Also unable to accept without a supplementary table of --entire-- transcriptional data. Only selected data is provided about author-selected genes. This selection is based on a "Custom script" used for analysis, unable to verify or repeat analysis. Unable to use data for future work if only subset chosen by unrepeatable analysis.

Please check Fig 5 -- if only a few microamps of current were produced from such large 65 cm^2 electrodes, the biofilms are barely alive and should not be used. It is very odd that the person who performed crucial experiements needed to produce the key data in this paper is not an author or part of any part of the analysis. If there is no author on this paper who understands the difference between good and bad growth on the electrode, it is suspect.

6. PLOS authors have the option to publish the peer review history of their article (what does this mean?). If published, this will include your full peer review and any attached files.

Reviewer #1: No

Reviewer #2: No

---

## [Author Response · Author response to Decision Letter 0]

20 Jul 2023

Reviewer #1: 

● The no of references should be reduced. 

Response: We have already reduced the number of them; however, the large number is due to the description of the genes differentially expressed, some with function studied in other bacteria.

The manuscript contains voluminous data that often confuses the reader as the objectives of the work are not mentioned clearly in the Introduction. However, the work is good and deserves publication after minor revision.

 Response: We add the aim of this work in the introduction.

Reviewer #2:

Appears to be an interesting paper following up on prior observations. The entire paper is based on transcriptional analysis, but the reviewer is unable to review it without access to RNAseq data. NCBI DEO database data is locked until 2025 without the secure token.

Response: Thanks, the data generated from the transcriptome analysis uploaded to the Gene Expression Omnibus repository has been released for inquiries and download. NCBI Gene Expression Omnibus database under accession number GSE223184.

No data on depth, % that was rRNA, or quality of RNA read data.

Response: Here are the data requested summarized in a table.

The table attached in the response letter

Also, unable to accept without a supplementary table of --entire-- transcriptional data. Only selected data is provided about author-selected genes. This selection is based on a "Custom script" used for analysis, unable to verify or repeat analysis. Unable to use data for future work if only subset chosen by unrepeatable analysis.

Response: The data of all the differentially expressed genes detected in the RNAseq analysis are reported in Supplementary Tables 2 and 3. 

Please check Fig 5 -- if only a few microamps of current were produced from such large 65 cm^2 electrodes, the biofilms are barely alive and should not be used. It is very odd that the person who performed crucial experiments needed to produce the key data in this paper is not an author or part of any part of the analysis. If there is no author in this paper who understands the difference between good and bad growth on the electrode, it is suspect.

Response: Thanks for the observation. KJ carried out the MFC experiments and Trevor L. Woodard supervised the work. Biofilms with viable cells were scraped (photos of the graphite electrode are attached).

 The low current data obtained may be due the application of a potential of 60 mV instead of 300 mV (normally used in these experiments). This decision was made because at the same time experiments were being carried out with FTO electrodes and we set the same conditions in order to compare both supports. In previous studies in our working group, we found that at 60 mV G. sulfurreducens biofilms formed on FTO electrodes reached limiting current in turn-over conditions in Cyclic Voltammetry experiments (Hernández-Eligio et al., 2022). We believe that this is the reason for the difference. As can be seen with respect to the WT strain there is an increase of 20%. On the other hand, these biofilms were collected and used for RNA extraction and RNA seq experiments (All quality data are adequate and indirectly reflect that the biofilms were active).

---

## [Editor Report · Decision Letter 1]

11 Oct 2023

Global transcriptional analysis of Geobacter sulfurreducens gsu1771 mutant biofilm grown on two different support structures

PONE-D-23-08621R1

Dear Dr. Juarez,

We’re pleased to inform you that your manuscript has been judged scientifically suitable for publication (necessary suggestions/concerned raised by the reviewers are addressed) and will be formally accepted for publication once it meets all outstanding technical requirements.

Kind regards,

Moupriya Nag

Academic Editor

PLOS ONE
---

## [Editor Report · Acceptance letter]

17 Oct 2023

PONE-D-23-08621R1 

Global transcriptional analysis of *Geobacter sulfurreducens gsu1771* mutant biofilm grown on two different support structures 

Dear Dr. Juarez:

I'm pleased to inform you that your manuscript has been deemed suitable for publication in PLOS ONE. Congratulations! Your manuscript is now with our production department. 

Kind regards, 

on behalf of

Dr. Moupriya Nag 

Academic Editor

PLOS ONE